# Tauroursodeoxycholic Acid Confers Protection Against Oxidative Stress via Autophagy Induction in Retinal Pigment Epithelial Cells

**DOI:** 10.3390/cimb47040224

**Published:** 2025-03-26

**Authors:** Daniella Zubieta, Cassandra Warden, Sujoy Bhattacharya, Milam A. Brantley

**Affiliations:** Department of Ophthalmology and Visual Sciences, Vanderbilt Eye Institute, Vanderbilt University Medical Center, Nashville, TN 37232, USA; daniella.zubieta@vumc.org (D.Z.); cwarden@oakland.edu (C.W.); sujoy.bhattacharya@vumc.org (S.B.)

**Keywords:** tauroursodeoxycholic acid, retinal pigment epithelial cells, autophagy, oxidative stress, age-related macular degeneration

## Abstract

Tauroursodeoxycholic acid (TUDCA) has been shown to protect against oxidative damage in retinal pigment epithelial (RPE) cells. However, the mechanisms by which it mediates these protective effects have not been thoroughly investigated in the context of age-related macular degeneration (AMD) disease onset and progression. We measured LC3-II and p62 expression via Western blot and immunohistochemistry in RPE cells treated with H_2_O_2_, TUDCA, or a combination of both to measure autophagy induction. To determine autophagy flux, we measured the expression of LC3-II/LC3-I in RPE cells in the presence of bafilomycin via Western blot. To determine the mechanistic pathways of TUDCA-induced autophagy, we measured the protein expression of autophagy regulators (Atg5, Beclin-1, S6, AMPK, and Akt) via Western blot. We show that TUDCA-mediated autophagy induction confers protection of RPE cells against oxidative damage via mTORC1/mTORC2 independent pathways but depends on Atg5. Our work adds to the overall understanding of RPE cell homeostasis and highlights the role of TUDCA in maintaining RPE health.

## 1. Introduction

Age-related macular degeneration (AMD) is the leading cause of irreversible blindness in older adults. In advanced stages, vision loss can result from geographic atrophy (GA), marked by the loss of photoreceptors and retinal pigment epithelium (RPE), or neovascular AMD (NVAMD), characterized by leakage from abnormal choroidal vessels [1]. Currently, there is limited treatment for GA, underscoring the need for ongoing research to develop better therapies and identify early intervention points to slow RPE degeneration. The lack of appropriate animal models that accurately mimic key hallmarks of atrophic AMD further complicates this challenge.

AMD is a multifactorial disease influenced by genetic and environmental factors. Genetic predispositions, such as variations in the complement factor H (CFH) gene, and environmental factors, including smoking and diet, play significant roles in its pathogenesis [2]. Oxidative stress is also a well-documented contributor, as RPE is highly susceptible to oxidative damage due to its high metabolic activity and constant exposure to light and oxygen [3]. This oxidative stress can lead to cellular dysfunction, inflammation, and cell death, exacerbating degeneration [3]. Understanding the interplay between these factors is crucial for developing comprehensive treatment approaches.

Bile acids, the major components of bile, are synthesized from cholesterol in the liver [4]. Bile acids are released into the duodenum upon food intake, where gut microbiota produce bile acids via hydroxylation, deconjugation, epimerization, and oxidation [4]. While bile acids are traditionally known for emulsifying lipids to promote their absorption during digestion, they are now recognized to modulate diverse metabolic processes such as energy, hepatic lipid, and glucose metabolism [5]. Beyond their established role in the gastrointestinal system, bile acids have emerged as potential modulators of retinal diseases.

Their interaction with bile acid receptors, such as the farnesoid X receptor (FXR) and the transmembrane G-protein-coupled receptor 5 (TGR5), has been implicated in the regulation of inflammation and cellular stress responses, positioning bile acids as promising candidates for ocular protection [6]. The bile acids ursodeoxycholic acid (UDCA) and tauroursodeoxycholic acid (TUDCA) protect against photoreceptor cell loss in multiple models of inherited retinal degeneration [7,8,9,10,11]. In addition, UDCA and TUDCA were found to protect against the development and progression of diabetic retinopathy in rodent models [12,13,14].

Few studies have examined bile acids’ role in RPE, a major cell type involved in AMD pathobiology. In an immortalized human RPE cell line, TUDCA has been shown to protect against oxidative damage [15]. Furthermore, TUDCA inhibited the proliferation of human retinal endothelial cells in vitro [16] and suppressed laser-induced choroidal neovascular membrane formation in rats [17].

Our earlier untargeted metabolomics studies demonstrated that plasma levels of glycine- and taurine-conjugated bile acids were altered in NVAMD patients compared to controls, suggesting that bile acid metabolism may be relevant to AMD pathophysiology [18,19]. Subsequently, we demonstrated that TCA inhibits features of AMD in vitro by preventing oxidative-stress-induced disruption of tight junctions in primary human RPE cells and suppressing VEGF-induced primary human choroidal endothelial cell migration and tube formation [20]. These data provide compelling evidence that bile acids confer protection for the RPE and choroid from AMD-related stimuli and suggest that employing bile acids or targeting bile acid receptors could transform treatments for GA and NVAMD. However, the mechanisms by which bile acids mediate these protective effects are incompletely understood and have not been thoroughly investigated in the context of AMD disease onset and progression. This knowledge gap is partly because no cell or animal models accurately replicate the key aspects of AMD pathophysiology. This limitation is due to the multifactorial nature of the condition, which complicates our understanding of the mechanisms involved in AMD pathogenesis.

Our investigations are grounded in the hypothesis that targeting bile acid pathways can confer RPE resilience against oxidative damage’s deleterious effects. We present data that reinforce the concept of bile acids as modulators of RPE survival and suggest that their therapeutic manipulation could offer a paradigm shift in the treatment of GA. Our data show that bile acid receptors are expressed in the human RPE cells, and bile acids can stimulate autophagy induction in both an immortalized RPE cell line, ARPE-19, and in primary human iPSC-derived RPE cells. We found that bile-acid-mediated autophagy flux activation confers protection of RPE cells against oxidative damage. Through a series of in vitro experiments, we elucidate the mechanisms by which bile acids exert their protective effects and discuss the implications of these findings for developing innovative therapies to prevent RPE degeneration in GA. We provide evidence for the role of bile acids in maintaining RPE health, suggesting their potential as a future treatment for AMD.

## 2. Materials and Methods

### 2.1. Cell Culture

Retinal pigment epithelial cells (ARPE-19) were purchased from ATCC (CRL-2302, Manassas, VA, USA) and were cultured in DMEM/F12 medium supplemented with 10% fetal bovine serum (FBS), 100 units/mL penicillin, and 100 μg/mL streptomycin. Before experimental procedures, ARPE-19 cells were grown in DMEM/F12 supplemented with 2% FBS, 100 units/mL penicillin, and 100 μg/mL streptomycin. RPE cells derived from human induced pluripotent stem cells (iPSC-derived RPE) were purchased from FujiFilm Cellular Dynamics Inc. (#R1102, Madison, WI, USA) and cultured in MEM alpha medium supplemented with 5% KnockOut SR (ThermoFisher #10828-28, Waltham, MA, USA), 1% N-2 supplement, 55 nM hydrocortisone, 250 µg/mL taurine, 14 pg/mL triiodo-l-thyronine, and 25 µg/mL gentamicin. The concentration of TUDCA (500 μM) used for experiments was determined from the dose curves of our previous studies [20,21]. All cultures were incubated at 37 °C, 5% CO_2_, 20.9% O_2_, and 95% relative humidity.

### 2.2. Quantitative PCR of Bile Acid Receptors in iPSC-RPE and ARPE-19 Cells

iPSC-derived RPE and ARPE-19 cells were grown to confluence, and total RNA was purified using the Qiagen RNEasy Mini Kit (74104, Germantown, MD, USA). RNA (2 µg) was reverse-transcribed using the High-Capacity cDNA Reverse Transcription Kit (Applied Biosystems, Waltham, MA, USA). Quantitative PCR was performed with six replicates for 60 cycles by co-amplification of TGR5 (Hs00544894_m1), FXR (Hs01026590_m1), PXR (Hs01114267_m1), and MCR (Hs01031804_m1) using gene-specific TaqMan Gene Expression Assays (Applied Biosystems, Waltham, MA, USA). Relative Ct values were averaged to represent the baseline expression of bile acid receptors.

### 2.3. MTT Assay

ARPE-19 cells were seeded onto 96-well microplates and grown to confluence. Cells were treated with 500 µM TUDCA, an H_2_O_2_ dose curve (200 µM, 400 µM, 600 µM), or a combination of these for 4 h. Untreated cells were used as a negative control. The Cell Proliferation Kit I (MTT) kit from Sigma (11465007001, St. Louis, MO, USA) was used and the instructions were followed. Briefly, 10 µL of the MTT labeling reagent was added to each well, and the microplate was incubated for 4 h. After incubation, 100 µL of the Solubilization buffer was added to each well and incubated overnight. Absorbance was measured using the SpectraMax M2 Plate Reader at 570 and 630 nm wavelengths. Triplicates of each sample were averaged, and the average absorbance value was normalized to the control.

### 2.4. Western Blots

ARPE-19 cells were seeded onto 6-well plates and grown to confluence. Cells were treated with 500 µM bile acid, 800 µM H_2_O_2_, or a combination of these for 24 h. We chose 800 µM as our concentration of H_2_O_2_ because we wanted cell death to be greater than 50%. According to the results of the MTT Assay, an H_2_O_2_ dose greater than 600 µM would be ideal for causing this cell death. Untreated cells were used as a negative control. After treatment, cells were washed once in cold PBS and stored overnight at −20 °C in 1x cell lysis buffer (Cell Signaling Technology, CST, Danvers, MA, USA) with 1:100 Halt protease and phosphatase inhibitor (ThermoFisher, Waltham, MA, USA). Cells were defrosted for 10 min on a shaker and centrifuged at 4 °C for 10 min at 10,000× *g* to collect supernatant. The total protein concentration of samples was determined using a BCA Assay (Pierce, ThermoFisher, Waltham, MA, USA), and 100 µg protein was concentrated using 10% trichloroacminetic acid for 15 min on ice. Samples were centrifuged at 15,000× *g* for 10 min, and the pellet was dissolved in Laemmli sample buffer (Bio-Rad, Hercules, CA, USA) with 5% BME (Sigma, St. Louis, MO, USA) overnight at 4 °C. Proteins were separated using 8%, 10%, 12%, or 15% acrylamide gels and electrophoretically transferred to polyvinylidene difluoride (PVDF) membranes (Millipore, Burlington, MA, USA). Membranes were blocked with 1% BSA in dd H_2_O and were probed with primary antibodies (p62 CST #5114, LC3A/B CST #12741, Atg5 CST #12994, Beclin-1 CST #3495, S6 CST #2317, p-S6 CST #2211, AMPK CST #2532, p-AMPK CST #2535, Akt CST #9272, p-Akt CST #9271) at 1:1000 dilution in 5% BSA in TBS + 0.1% Tween20 (TBS-T) overnight at 4 °C. Membranes were washed 3 times with TBS-T, and secondary antibody goat anti-rabbit IgG (Millipore #12-348) in 5% milk in TBS-T was added to the blots for 1 h at room temperature on a shaker. After washing the membranes 3 times with TBS-T, the chemiluminescence substrate ECL (Pierce, ThermoFisher, Waltham, MA, USA) was added for development, and the membranes were imaged using the ChemiDoc MP Imaging System (Bio-Rad, Hercules, CA, USA). Band intensity was quantified using ImageJ V1.53t. Beta-actin (CST #12262) was used as a loading control.

For the autophagy flux experiments, ARPE-19 and iPSC-derived RPE cells were treated with 500 µM TUDCA, 100 nM bafilomycin, or a combination of these for 3 h. The Western blot protocol was followed as above.

### 2.5. Immunocytochemistry in iPSC-Derived RPE and ARPE-19 Cells

iPSC-derived RPE and ARPE-19 cells were seeded onto chambered well slides at 50% confluence and treated with 500 µM TUDCA, 200 µM H_2_O_2_, or a combination of these for 16 h. Sparse cells were used because they can provide spatial and morphological details about autophagosomes, lysosomes, and their co-localization at the single-cell level. We chose 200 µM as our concentration of H_2_O_2_ because we did not want to cause cell death. Instead, we just wanted a low dose of H_2_O_2_ to show the colocalization of this treatment with TUDCA. Cells were fixed with 1:1 methanol/acetone for 10 min and blocked in 2% BSA in PBS + Tween20 (0.05%) for 1 h. Primary antibodies LC3 (1:200, CST #12741) and p62 (1:200, CST #88588) were added to the blocking buffer for 1 h at room temperature. Following three washes with PBS, secondary antibodies were added to the blocking buffer for 1 h at room temperature, protected from light. Slides were imaged using an Olympus FV1000 confocal microscope. Images were analyzed using ImageJ V1.53t (National Institutes of Health, Bethesda, MD, USA) to measure the relative intensity of each colocalization.

For the bile acid receptor staining experiments, untreated ARPE-19 was used. The Immunocytochemistry protocol was followed as above with the primary antibodies FXR (1:50, Invitrogen PA5-40755, Waltham, MA, USA) and PXR (1:100, Invitrogen MA5-31808, Waltham, MA, USA).

### 2.6. Statistical Analysis

Values of assay replicates were averaged for each assay, and standard deviation was calculated. Statistically significant differences were determined using one-way ANOVA with Tukey Honest Significant Differences post hoc analysis in R V4.4.1. *p* < 0.05 was considered significant.

## 3. Results

### 3.1. Detection and Baseline Expression of Bile Acid Receptors in RPE Cells

Given that our earlier untargeted metabolomics studies demonstrated that plasma levels of certain bile acids were altered in NVAMD patients, we hypothesized that bile acids could confer RPE resilience against oxidative damage. To investigate this, we first sought to determine if bile acid receptors were expressed in ARPE-19 and iPSC-derived RPE cells. Using qRT-PCR to measure baseline gene expression of cytoplasmic receptors TGR5 and PXR and nuclear receptors MCR and FXR (Appendix A), we found moderate expression of MCR (Ct = 25.654) and PXR (Ct = 32.289) and low expression of FXR (Ct = 36.917) and TGR5 (Ct = 36.806) in ARPE-19 cells. When we measured the expression of these receptors in iPSC-derived RPE cells, we found moderate expression of MCR (Ct = 30.284), very low expression of TGR5 (Ct = 52.042), and no expression of FXR or PXR. We also detected relative protein expression of bile acid receptors in ARPE-19 cells via Western blot (Appendix A), and localization of the receptors via fluorescent staining (Appendix A). We found that PXR and TGR5 had moderate protein expression, but FXR had a low expression, confirming that bile acid receptors are detectable in RPE cells.

### 3.2. TUDCA Protects Against H_2_O_2_-Induced Cell Death

Alhasani et al. [15] reported that TUDCA is protective against H_2_O_2_-induced cell death in ARPE-19 cells. We used a dose curve to determine the amount of H_2_O_2_ needed to cause oxidative-stress-induced cell death in our APRE-19 cells (Figure 1). Using an MTT assay, we found no changes in cell viability in ARPE-19 cells treated with 200 µM H_2_O_2_ (*p* = 0.89) and a decrease in cell viability at 400 µM (*p* = 0.037) and 600 µM H_2_O_2_ (*p* = 1.8 × 10^−4^) compared to untreated controls. These data suggest that 400 µM H_2_O_2_ or higher doses cause cell death in ARPE-19 cells. Adding 500 µM TUDCA prevented H_2_O_2_-induced cell death at both 400 µM (*p* = 0.036) and 600 µM (*p* = 0.016) compared to cells treated with H_2_O_2_ alone, confirming that TUDCA can prevent ARPE-19 cell death at toxic levels of oxidative stress.

### 3.3. TUDCA Initiates RPE Autophagy by Increasing LC3-II Expression and LC3/p62 Colocalization

Previous studies have shown that RPE cells from patients with advanced AMD exhibit impaired autophagy compared to RPE from age-matched healthy controls [22,23]. This suggests that dysfunctional RPE autophagy may be a contributing factor in AMD. Because bile acids have been shown to increase autophagy in other epithelial cell types [24,25], we wanted to determine if TUDCA could protect against oxidative stress by inducing autophagy in RPE cells. We investigated this by focusing on LC3 and p62 proteins, which are autophagic markers. Under basal conditions, LC3 is localized in the cytoplasm as LC3-I. LC3 is translocated to autophagosome membranes during autophagy, which converts LC3-I into LC3-II [26]. Thus, we expect an accumulation of LC3-II in cells undergoing autophagy because this indicates active autophagosome formation. Through the basal autophagy processing, p62 is continuously cleared from the cytoplasm [27]. Its levels are inversely related to autophagic flux, and therefore, we would expect a decrease in the levels of p62 in cells undergoing autophagy.

To measure autophagy induction, we measured the protein expression of LC3 and p62 in ARPE-19 cells. We found no changes in LC3-II in cells treated with H_2_O_2_ (*p* = 0.70) or TUDCA (*p* = 0.69) alone (Figure 2A), but an increase in LC3-II when added together compared to untreated controls (*p* = 5.3 × 10^−5^). These data imply that TUDCA induces autophagy in the presence of oxidative stress. We also saw a decrease in p62 expression in cells treated with H_2_O_2_ (*p* = 0.049) and TUDCA (*p* = 0.047) alone (Figure 2B). When H_2_O_2_ and TUDCA were added together, p62 levels were nominally lower, but the change was not statistically significant (*p* = 0.20). These data imply that autophagy is highly induced when RPE cells are exposed to H_2_O_2_ and TUDCA. While the induction is not to the same degree, autophagy is still increased when TUDCA is used in the presence of oxidative stress.

To further measure autophagy, we directly observed the colocalization of LC3 and p62 via immunocytochemistry [28]. The colocalized structures, based on the appearance of puncta, would correspond to autophagosomes [29], and suggest the induction of autophagy. We found an increase in LC3 and p62 colocalization in ARPE-19 cells treated with H_2_O_2_ alone (*p* < 1. 0 × 10^−7^), TUDCA alone (*p* = 0.035), and H_2_O_2_ + TUDCA (*p* < 1.0 × 10^−7^) compared to untreated controls (Figure 3). These data further suggest that TUDCA initiates autophagy in response to oxidative stress in RPE cells.

### 3.4. TUDCA Increases Autophagy Flux in the Presence of Bafilomycin

An increase in LC3-II levels could indicate either enhanced autophagosome formation or a blockage in the later stages of autophagy [28]. Autophagy flux can be used to differentiate between these two scenarios. Flux is more accurately represented by LC3-II differences between samples in the presence of an inhibitor, such as bafilomycin [30]. If we were to see a further accumulation of LC3-II, this would indicate an increase in autophagy flux since bafilomycin inhibits autophagosome degradation [30,31].

For the autophagy flux experiments, we measured LC3-II expression in ARPE-19 cells treated with bafilomycin (Figure 4A). There was no change in LC3-II expression in ARPE-19 cells treated with TUDCA alone (*p* = 0.99). Bafilomycin alone (*p* = 4.7 × 10^−4^) and bafilomycin + TUDCA (*p* = 2.5 × 10^−4^) increased LC3-II expression compared to untreated controls. Furthermore, there was no change in LC3-II expression in ARPE-19 cells treated with bafilomycin + TUDCA compared to bafilomycin alone (*p* = 0.16).

To investigate autophagy flux further, we measured the LC3-II/LC3-I ratio since this would reflect the conversion of the protein from the cytosolic form to the autophagosome form following autophagy flux [32]. The LC3-II/LC3-I ratios showed that there was no change in LC3-I conversion in ARPE-19 cells treated with TUDCA alone (*p* = 0.93), and there was a significant increase in LC3-I conversion to LC3-II in cells treated with bafilomycin alone (*p* = 0.0019) and bafilomycin + TUDCA (*p* = 4.0 × 10^−4^) compared to untreated controls (Figure 4B). We also found an increase in the LC3-II/LC3-I ratios in cells treated with bafilomycin + TUDCA compared to bafilomycin alone (*p* = 0.026). These data imply that the accumulation of LC3-II in cells treated with TUDCA and bafilomycin is due to autophagy flux.

### 3.5. TUDCA Initiates RPE Autophagy Through Atg Proteins

Next, we sought to determine the signaling pathway of TUDCA-initiated autophagy by measuring the protein expression of autophagy regulators. Autophagy protein 5 (Atg5) is necessary for autophagic vesicle formation [33]. Beclin-1 is another crucial regulator of autophagy and plays a significant role in the initiation and regulation of the autophagic process [34]. We investigated these autophagic markers to determine if TUDCA activates autophagy through these proteins. We found an increase in Atg5 protein levels in ARPE-19 cells treated with TUDCA alone (*p* = 9.2 × 10^−3^) and H_2_O_2_ + TUDCA (*p* = 3.6 × 10^−3^) compared to untreated controls (Figure 5B), but no difference in cells treated with H_2_O_2_ alone (*p* = 0.35). These findings suggest that TUDCA induces autophagy through Atg proteins. There was no difference in Beclin-1 expression in cells treated with H_2_O_2_ alone (*p* = 0.52), TUDCA alone (*p* = 0.75), or H_2_O_2_ + TUDCA (*p* = 0.52) compared to untreated controls (Figure 5C). These observations suggest that TUDCA-induced autophagy is Beclin-1-independent.

The mammalian target of rapamycin (mTOR) controls autophagy by inhibiting the process in the presence of abundant nutrients [34]. There are two main complexes of mTOR: mTORC1 and mTORC2 [35]. AMP-activated protein kinase (AMPK) promotes autophagy by inhibiting mTORC1 and phosphorylating ULK1 [34,36]. Since Ribosomal protein S6 is activated downstream of mTORC1 [37], we used changes in AMPK and S6 expression to determine if TUDCA activates mTORC1. There was a decrease in ribosomal S6 activation in RPE cells with H_2_O_2_ alone (*p* = 0.018) and H_2_O_2_ + TUDCA (*p* = 0.010) compared to untreated controls (Figure 5D), but no difference with TUDCA alone (*p* = 0.17), suggesting that TUDCA does not activate the mTORC1 complex. ARPE-19 cells treated with H_2_O_2_ alone showed an increase in AMPK activation (Figure 5E) compared to untreated controls (*p* = 0.029), and this increase was inhibited in ARPE-19 cells treated with TUDCA compared to H_2_O_2_ alone (*p* = 0.016). TUDCA alone had no significant changes in AMPK activation (*p* = 0.86), suggesting that TUDCA is AMPK-independent and does not activate mTORC1. These data imply that the signaling pathway of TUDCA initiation of autophagy is mTORC1-independent.

Because we saw that TUDCA does not activate mTORC1, we investigated mTORC2 activity through Akt expression. Increased phosphorylation of Akt at Ser473 indicates active mTORC2 [38]. We found no changes in Akt activity in ARPE-19 cells treated with H_2_O_2_ alone (*p* = 0.23) or TUDCA alone (*p* = 0.99) compared to untreated cells (Figure 6), but we found an increase in Akt activity in cells treated with H_2_O_2_ + TUDCA compared to untreated controls (*p* = 6.3 × 10^−5^). TUDCA alone does not affect Akt, which suggests that the signaling pathway of TUDCA-initiation of autophagy is mTOR2-independent. H_2_O_2_ induces oxidative stress, activating various signaling pathways, including those involved in autophagy. Therefore, combining TUDCA and H_2_O_2_ might create a cellular environment that favors autophagy through pathways that do not include mTORC2, potentially explaining the observed effect.

### 3.6. TUDCA Activates Autophagy in iPSC-Derived RPE Cells

ARPE-19 cells have been used in vision research for decades because they share many structural and functional properties with RPE cells in vivo [39]. However, this may not always be the case, and therefore, we needed to confirm our findings in cells that resembled primary RPE and were non-immortalized [40]. Therefore, to confirm our ARPE-19 findings, we tested if TUDCA increases LC3/p62 colocalization in iPSC-derived RPE cells (Appendix A, Figure 7). We found TUDCA alone (*p* = 6.7 × 10^−6^), H_2_O_2_ alone (*p* = 1.0 × 10^−7^), and H_2_O_2_ + TUDCA (*p* = 1.0 × 10^−7^) increased LC3/p62 localization compared to untreated controls. This observation is consistent with our ARPE-19 data, showing that TUDCA initiates autophagy in RPE cells.

Furthermore, we wanted to confirm that TUDCA activates autophagy in iPSC-derived RPE cells via autophagy flux experiments. First, we measured LC3-II expression in cells treated with bafilomycin (Figure 8A). There was no change in LC3-II expression in iPSC-derived RPE cells treated with TUDCA alone (*p* = 0.92) or bafilomycin + TUDCA (*p* = 0.056). Bafilomycin alone (*p* = 0.040) increased LC3-II expression compared to untreated controls.

Next, we measured the LC3-II/LC3-I ratio. The LC3-II/LC3-I ratios showed that there was no change in LC3-I conversion in iPSC-derived RPE cells treated with TUDCA alone (*p* = 1.0), and there were no significant increases in LC3-I conversion to LC3-II in cells treated with bafilomycin alone (*p* = 0.205) and bafilomycin + TUDCA (*p* = 0.058) compared to untreated controls (Figure 8B). These iPSC-derived RPE data show a trend that aligns with the ARPE-19 data but requires further validation. While statistical significance was not reached in this dataset, the biological relevance of the observed pattern remains strong, suggesting that future work with increased sample sizes or optimized differentiation protocols may help clarify this further.

### 3.7. RPE Autophagy Induction by Other Bile Acids

To test whether autophagy induction is unique to TUDCA or can be seen with other bile acids, we tested LC3 and p62 levels in ARPE-19 cells treated with taurocholic acid (TCA) and taurochenodeoxycholic acid (TCDCA). We chose TCA because our previous study showed that TCA protected RPE tight junctions from oxidative stress [20], and TCDCA acted as an additional hydrophobic, taurine-conjugated bile acid [41].

We found no changes in LC3-II expression in cells treated with TCA (*p* = 1.0) or TCDCA (*p* = 0.93) alone compared to untreated controls (Figure 9A). LC3-II expression increased in cells treated with H_2_O_2_ + TCA (*p* = 0.0093) and H_2_O_2_ + TCDCA (*p* = 0.037) compared to controls. There was no difference in p62 levels in cells treated with TCA alone (*p* = 0.93), TCDCA alone (*p* = 0.99), or H_2_O_2_ + TCA (*p* = 0.41) compared to controls, but an increase was found in cells treated with H_2_O_2_ + TCDCA compared to untreated controls (*p* = 0.0026) (Figure 9B). These data suggest that autophagy induction could be unique to TUDCA, but further studies are needed.

## 4. Discussion

Our studies demonstrate for the first time that a naturally produced hydrophilic bile acid, TUDCA, regulates autophagy in RPE cells, providing critical insights into RPE biology during aging and pathology. We present novel in vitro evidence that bile acid-mediated autophagy activation protects RPE cells against oxidative-stress-induced damage. TUDCA’s ability to induce autophagy flux was observed in both ARPE-19 cells, a human RPE cell line commonly used for research [42], and iPSC-derived RPE cells, which are derived from pluripotent stem cells and more closely mimic the native RPE’s characteristics [43]. However, iPSC-derived RPE and ARPE-19 cells have different genetic and epigenetic profiles, which may affect their response to treatments and experimental conditions. Also, iPSC-derived RPE cells may not be fully mature or may exhibit variability in their differentiation state, which can influence experimental outcomes. The differences in responses between ARPE-19 and iPSC-RPE cells can be attributed to several factors. iPSC-RPE cells closely resemble native RPE in terms of morphology, gene expression, and functionality, while ARPE-19 cells are more dedifferentiated and may not fully replicate native RPE behavior. iPSC-RPE cells typically exhibit more robust autophagic and lysosomal activity and a more physiologically relevant polarized monolayer, which could influence their responses to stressors and treatments. Additionally, variability in differentiation protocols for iPSC-RPE cells and differences in baseline stress responses between the two models may contribute to the observed discrepancies. These factors highlight the need for caution when interpreting data across different RPE cell models. This is a major limitation of this study, and additional experiments are warranted. Despite this, the fact that both ARPE-19 and iPSC-derived RPE cells exhibit similar responses regarding autophagy regulation highlights the robustness of TUDCA’s protective effects across different RPE cell models. These observations validate our findings and underscore the potential of using TUDCA as a therapeutic agent for conditions such as atrophic AMD.

We began by determining the expression of bile acid receptors in RPE cells. Previous studies have focused on these receptors in retinal endothelial cells [14,44]. This study presents unique findings that FXR, MCR, PXR, and TGR5 bile acid receptors are present in RPE cells. Knowing that bile acid receptors are present in RPE cells could reveal new aspects of their regulatory mechanisms. The presence of bile acid receptors could be linked to various diseases. For example, dysregulation of these receptors might contribute to AMD or diabetic retinopathy. Understanding these connections could lead to new therapeutic targets. Bile acids are involved in various metabolic pathways, including lipid metabolism and inflammation [5]. The presence of these receptors in RPE cells suggests that these pathways might also be critical in the retina, influencing health and disease. Finally, understanding the role of bile acid receptors in RPE cells adds to the broader knowledge of cell biology and physiology, contributing to our overall understanding of how RPE cells maintain homeostasis.

While we found MCR and TGR5 expression in both cell types, there was a difference in FXR and PXR expression between iPSC-derived RPE and ARPE-19 cells. This difference can be attributed to their distinct origins, culture conditions, and genetic/epigenetic modifications. iPSC-derived RPE cells may not fully express these transcription factors due to incomplete differentiation or specific culture conditions, whereas ARPE-19 cells, being an immortalized line, may retain or overexpress these genes due to their unique genetic makeup and culture environment. Understanding these differences is crucial for selecting appropriate cell models for specific research applications and interpreting experimental results accurately.

An essential part of our experimental design focused on TUDCA’s ability to protect against H_2_O_2_-induced cell death in RPE cells. We found that cells treated with 500 µM of TUDCA protected RPE cells treated as high as 600 µM H_2_O_2_. These findings are consistent with an earlier report by Alhasani et al. [15] and provide a strong rationale for exploring the mechanisms by which TUDCA exerts its protective effects. TUDCA has been shown to alleviate the endoplasmic reticulum (ER) stress under serum starvation [45], tunicamycin-induced apoptosis [46], and ferroptosis [47]. In the retina, TUDCA protects photoreceptors from cell death after experimental retinal detachment [48], in vivo models of retinal disorders [49], and oxidative-stress-induced retinal degeneration [50]. These studies provide evidence that TUDCA can delay the degeneration of retinal neurons and preserve retinal structure and function through its ability to inhibit apoptosis, attenuate oxidative stress, decrease inflammation, reduce ER stress, and pathological angiogenesis [49,51], suggesting that TUDCA is a potent neuroprotective agent.

Previous studies have investigated the role of TUDCA in autophagy in different systems [52,53]. The current study is unique because we examined the effects of TUDCA-induced autophagy in both immortalized ARPE-19 and primary iPSC-derived RPE cells in vitro. We provide evidence that TUDCA alone and in the presence of H_2_O_2_-induced oxidative stress induces autophagy in RPE cells. While the expression of LC3-II does not increase with TUDCA alone, it does show an increase when TUDCA is combined with H_2_O_2_. This observation suggests that TUDCA does not stimulate autophagy unless the cells undergo oxidative stress. Both H_2_O_2_ and TUDCA individually decrease p62 expression, indicating increased autophagic activity. However, the p62 levels do not change when TUDCA and H_2_O_2_ are added together. This could imply several possibilities, including saturation of the autophagy pathway due to maximal activation by either H_2_O_2_ or TUDCA alone. The combined treatment might activate or inhibit alternative pathways that modulate p62 levels, resulting in no net change in p62 expression. The colocalization results also support the conclusion that TUDCA induces autophagy due to the increased puncta in RPE cells treated with TUDCA alone and combined with H_2_O_2_. These puncta correspond with autophagosomes, suggesting autophagy activity.

Because TUDCA stimulated an increase in LC3-II/LC3-I in the presence of an inhibitor, we can conclude that TUDCA enhances autophagosome formation instead of impairing degradation in the later stages of autophagy. This effect is novel evidence that suggests TUDCA enhances autophagy flux in RPE cells. Since autophagy is a cellular process that involves the degradation and recycling of damaged organelles and proteins [54], we posit that TUDCA can protect RPE cells from stress-induced damage through autophagy stimulation, which is crucial for maintaining RPE cell homeostasis and function. To clarify whether autophagy mediates this protection, experiments using autophagy inhibitors, such as 3-MA or chloroquine, could determine if blocking autophagy reduces TUDCA’s efficacy. However, due to limitations in time and resources, it is not feasible to confirm causation through further experiments at this stage. We acknowledge this constraint and propose that future studies investigate whether inhibiting autophagy impacts TUDCA’s protective role. TUDCA’s ability to stimulate autophagy suggests it could be a valuable therapeutic agent for treating retinal diseases where impaired autophagy and RPE dysfunction play a critical role, such as atrophic AMD.

Earlier work by Murase et al. [55] showed that TUDCA activates phagocytosis in RPE cells through an independent pathway of ER stress. RPE cells play a major role in maintaining photoreceptor health by phagocytosing the shed photoreceptor outer segments (POSs). TUDCA was shown to activate the Mer tyrosine kinase receptor (MerTk) in RPE cells, which is essential for the phagocytosis of POSs. Most MerTk mutations in patients are associated with early-onset retinal dystrophy, so the observation that TUDCA increases MerTk activity can have implications for RPE health and survival. Although the mechanisms by which TUDCA increases MerTk activity are unclear, it appears that TUDCA-mediated integrin activation can contribute to MerTk activity [55]. LC3-associated phagocytosis (LAP) in RPE represents a crucial non-canonical hybrid pathway, typically associated with autophagy but distinct in its mechanism [56]. Unlike autophagy, LAP is AMPK-mTORC1-independent and unresponsive to nutrient status [57]. In RPE cells, LAP plays a vital role in the clearance of POSs, thereby preventing the accumulation of toxic byproducts [58]. Given that bile acids regulate both RPE canonical autophagy and phagocytosis, it is reasonable to speculate that they could also play a role in LAP in maintaining retinal health and protecting against retinal degenerative diseases.

Our work provides novel evidence of the mechanistic pathways that TUDCA uses to induce RPE autophagy. We found that TUDCA activates autophagy flux via mTORC1/mTORC2 independent pathways but depends on Atg5. Conversely, H_2_O_2_ induces RPE autophagy through mTORC1 and AMPK-dependent pathways. Our results also suggest that TUDCA acts differently in the presence of H_2_O_2_-oxidative stress, which we expect is due to its protection of RPE cells. Interestingly, TUDCA inhibits H_2_O_2_-induced AMPK activation but increases H_2_O_2_-induced Akt activation, underscoring its complex role in modulating cellular responses to oxidative stress. TUDCA may reduce AMPK activation induced by oxidative damage to promote RPE cell viability. On the other hand, TUDCA has been shown to suppress the mTOR pathway in hepatocytes, potentially contributing to its protective effects against liver injury [59]. Other studies also show that TUDCA’s beneficial effects are AMPK-dependent, primarily in maintaining energy and glucose homeostasis [60]. Our studies show that TUDCA’s cytoprotective effect in RPE cells is AMPK-independent. The activation of the Akt pathway by TUDCA in the presence of OS has significant implications for both cytoprotection and autophagy induction. Akt can phosphorylate and inhibit several pro-apoptotic factors, thereby enhancing cell survival under stress conditions. Akt activation can also induce autophagy. While Akt is typically associated with the mTOR pathway, which inhibits autophagy, there are situations where Akt activation can promote autophagy indirectly. For instance, Akt can modulate the activity of transcription factors like FOXO [61], which can upregulate autophagy-related genes. In summary, TUDCA activation of Akt in the presence of oxidative stress could enhance cytoprotection by inhibiting apoptotic pathways and potentially promoting autophagy through alternative signaling routes. Thus, it is crucial to consider that the specific cellular context, the type of stressor, and the duration of TUDCA treatment can diversely influence these signaling pathways.

We investigated whether RPE autophagy induction was seen with other bile acids. The lack of change in LC3-II levels in cells treated with TCA or TCDCA alone suggests that these treatments do not significantly affect autophagosome formation or the autophagic process under the conditions tested. The increase in LC3-II levels with the combined treatments of H_2_O_2_ and TCA or TCDCA indicates an accumulation of autophagosomes. This accumulation could be due to increased autophagosome formation or a blockage in autophagosome–lysosome fusion, leading to their accumulation. However, the concurrent increase in p62 levels indicates that these autophagosomes are not efficiently degraded, indicating impairment in autophagic flux. We cannot be sure of this conclusion without repeating an autophagy flux experiment with TCA and TCDCA, which is a limitation of this study. The lack of conclusive evidence could be a relevant next step in determining if autophagy induction is unique to TUDCA or can be seen with other bile acids.

This study shows that TUDCA is a protective agent against oxidative stress through RPE autophagy induction. Our work not only adds to the overall understanding of RPE cell homeostasis but also highlights the role of TUDCA in maintaining RPE health, which suggests its potential to transform the therapeutic landscape of AMD.

## Figures and Tables

**Figure 1 cimb-47-00224-f001:**
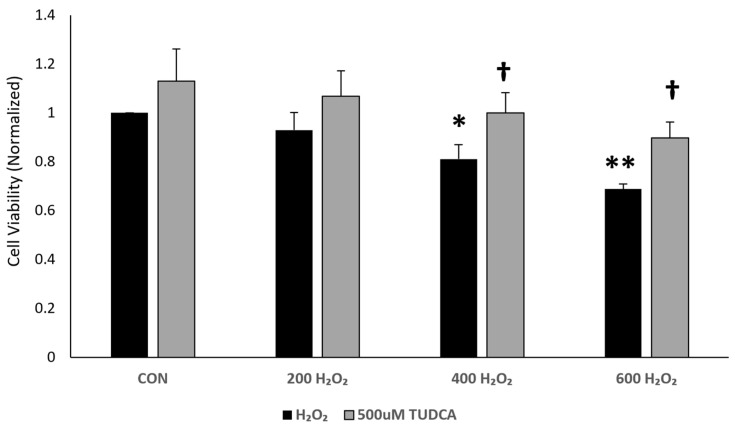
TUDCA protects against H_2_O_2_-induced cell death. ARPE-19 cells were treated with increasing doses of H2O2 (200 µM, 400 µM, and 600 µM), 500 µM TUDCA, or H_2_O_2_ + TUDCA. Untreated cells served as a control. Cell viability results via an MTT Assay (n = 4); normalized to control. * *p* < 0.05 compared to control; ** *p* < 0.001 compared to control; † *p* < 0.05 compared to H_2_O_2_.

**Figure 2 cimb-47-00224-f002:**
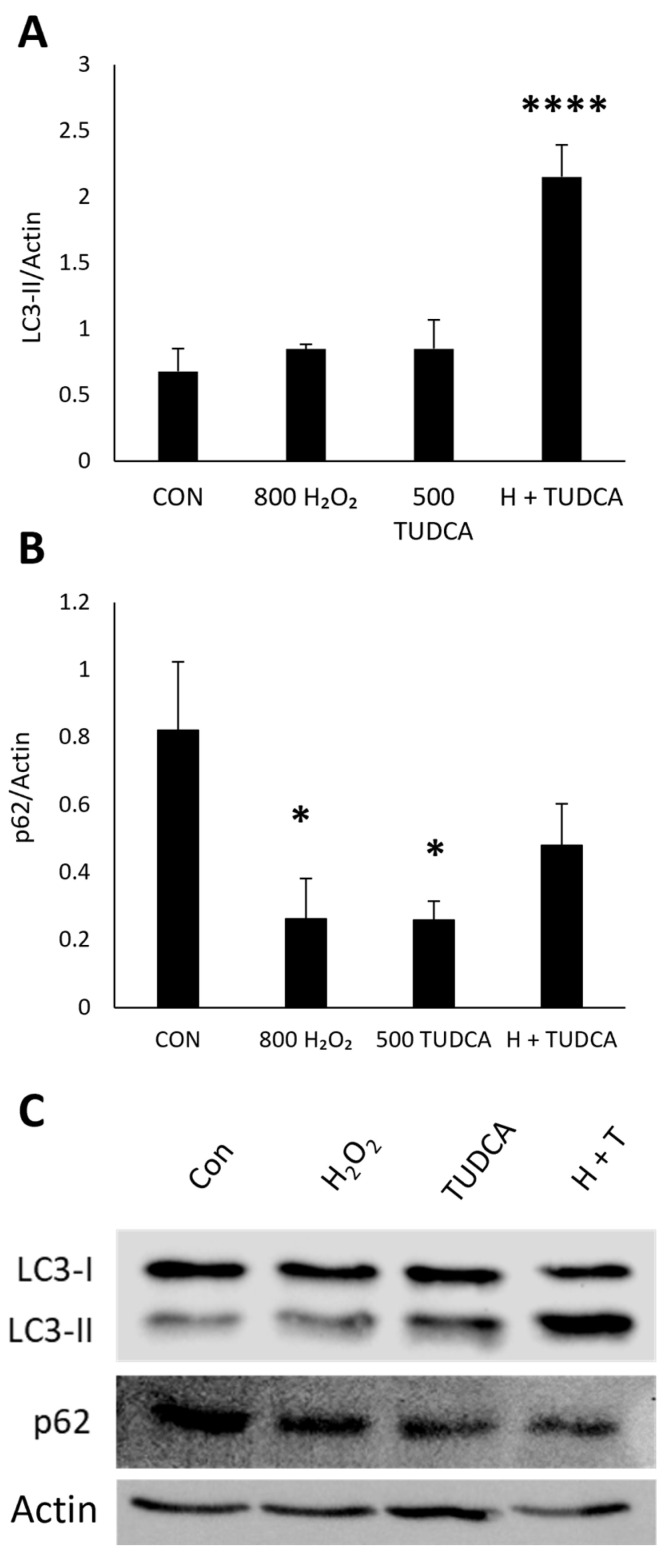
TUDCA increases LC3-II protein expression. ARPE-19 cells were treated with 800 µM H_2_O_2_, 500 µM TUDCA, or H_2_O_2_ + TUDCA for 24 h, and protein lysate was collected to measure LC3 and p62 expression. Untreated cells served as a control. Data of (**A**) LC3-II (n = 3) and (**B**) p62 (n = 2) protein expression represented as mean ± standard deviation. * *p* < 0.05 compared to control. **** *p* < 0.0001 compared to control. (**C**) Representative Western blots of LC3 and p62.

**Figure 3 cimb-47-00224-f003:**
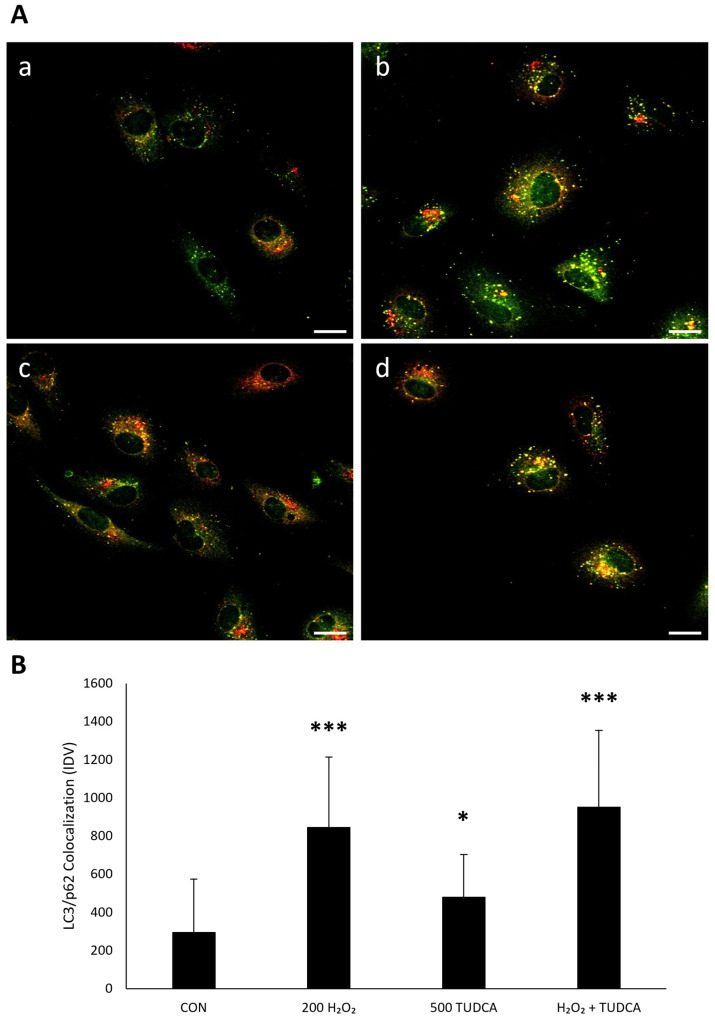
TUDCA and H_2_O_2_ increase the colocalization of LC3 and p62. (**A**) ARPE-19 cells were (**a**) untreated; treated with (**b**) 200 µM H_2_O_2_, (**c**) 500 µM TUDCA, or (**d**) H_2_O_2_ + TUDCA for 24 h and were stained for LC3 and p62 localization. (**B**) Quantification of colocalized LC3 and p62. * *p* < 0.05 *** *p* < 0.001 compared to control. Green puncta: LC3. Red puncta: p62. Yellow puncta: colocalization. Magnification, 60×. Scale bar, 20 µm.

**Figure 4 cimb-47-00224-f004:**
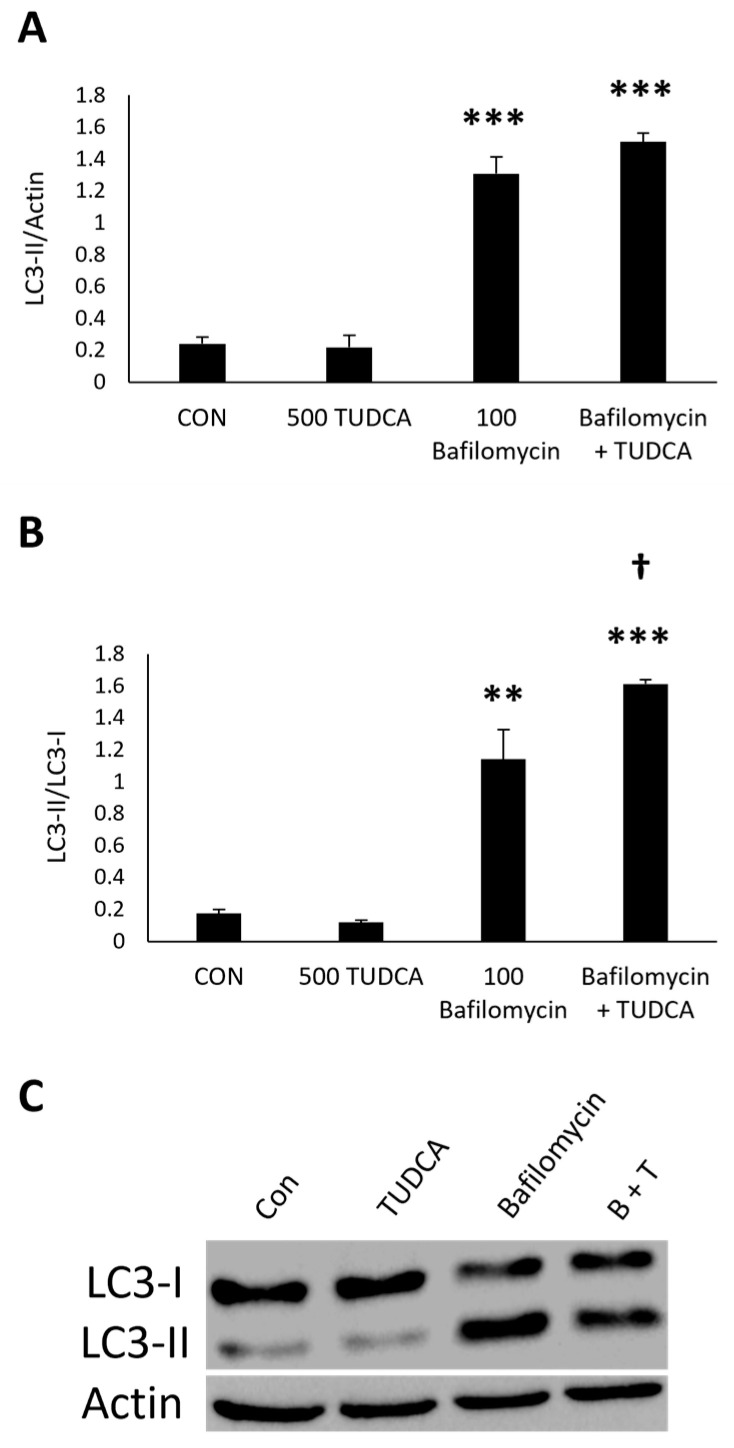
TUDCA increases autophagy flux. ARPE-19 cells were treated with 500 µM TUDCA, 100 nM bafilomycin, or bafilomycin + TUDCA for 3 h, and protein lysate was collected to measure LC3 expression. Untreated cells served as a control. Data of LC3-II (**A**) and LC3-II/LC3-I (**B**) protein expression represented as mean ± standard deviation (n = 2). ** *p* < 0.01 compared to control. *** *p* < 0.001 compared to control. † *p* < 0.05 compared to bafilomycin alone. (**C**) Representative Western blots of LC3.

**Figure 5 cimb-47-00224-f005:**
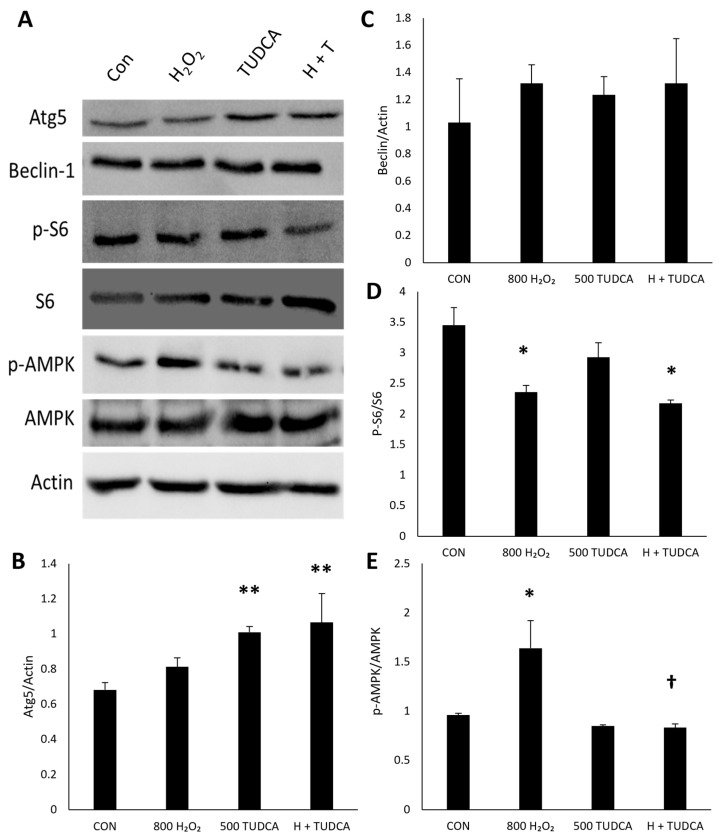
TUDCA increases Atg5 expression. (**A**) Representative blots of autophagy regulation proteins. Data of (**B**) Atg5 (n = 3), (**C**) Beclin-1 (n = 3), (**D**) phosphorylated S6/total S6 (n = 2), and (**E**) phosphorylated AMPK/total AMPK (n = 2) are presented as mean ± standard deviation. * *p* < 0.05 compared to control. ** *p* < 0.01 compared to control; † *p* < 0.05 compared to H_2_O_2_.

**Figure 6 cimb-47-00224-f006:**
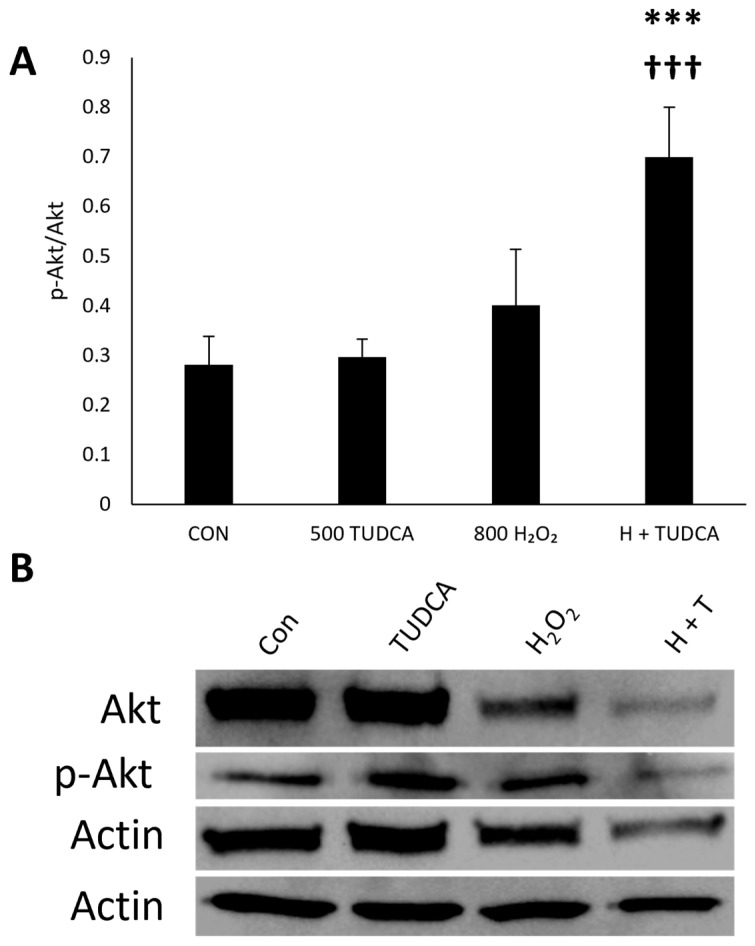
TUDCA and H2O2 activate Akt. Data of (**A**) phosphorylated Akt/total Akt are presented as mean ± standard deviation (n = 4). (**B**) Representative blots of autophagy regulation proteins. *** *p* < 0.0001 compared to control; ††† *p* < 0.0001 compared to TUDCA alone.

**Figure 7 cimb-47-00224-f007:**
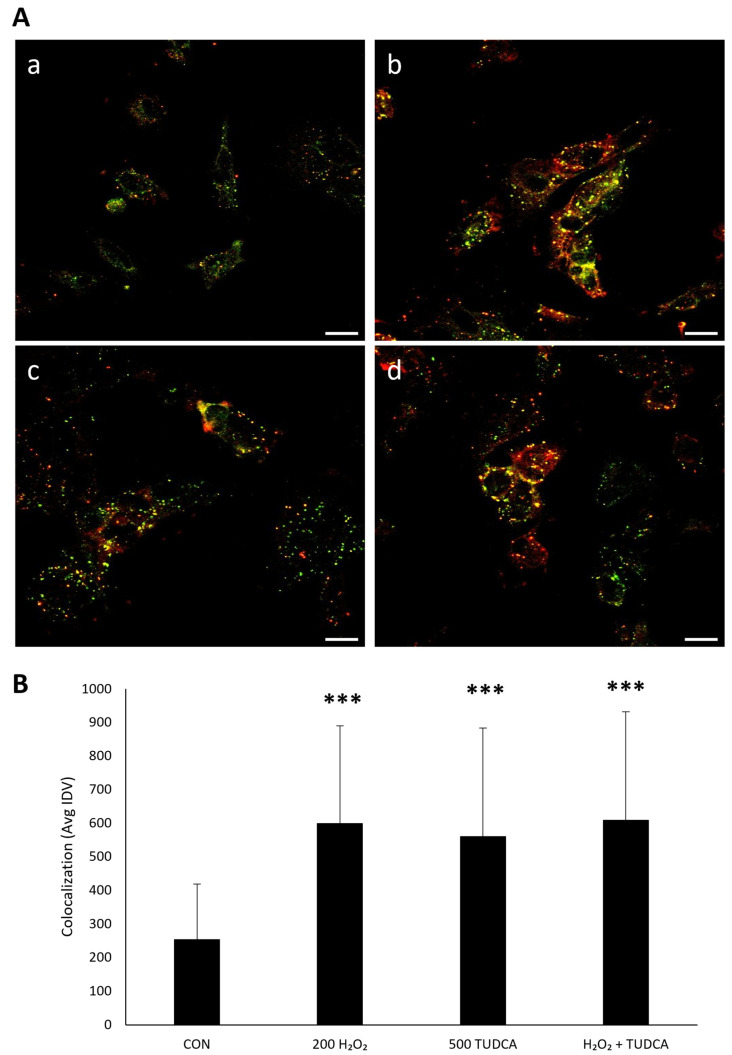
TUDCA and H_2_O_2_ increase the colocalization of LC3 and p62 in iPSC-derived RPE. (**A**) iPSC-derived RPE cells were (**a**) untreated, treated with (**b**) 200 µM H_2_O_2_, (**c**) 500 µM TUDCA, or (**d**) H_2_O_2_ + TUDCA for 24 h and were stained for LC3 and p62 localization. (**B**) Quantification of colocalized LC3 and p62. *** *p* < 0.001 compared to control. Green puncta: LC3. Red puncta: p62. Yellow puncta: colocalization. Magnification, 60×. Scale bar, 20 µm.

**Figure 8 cimb-47-00224-f008:**
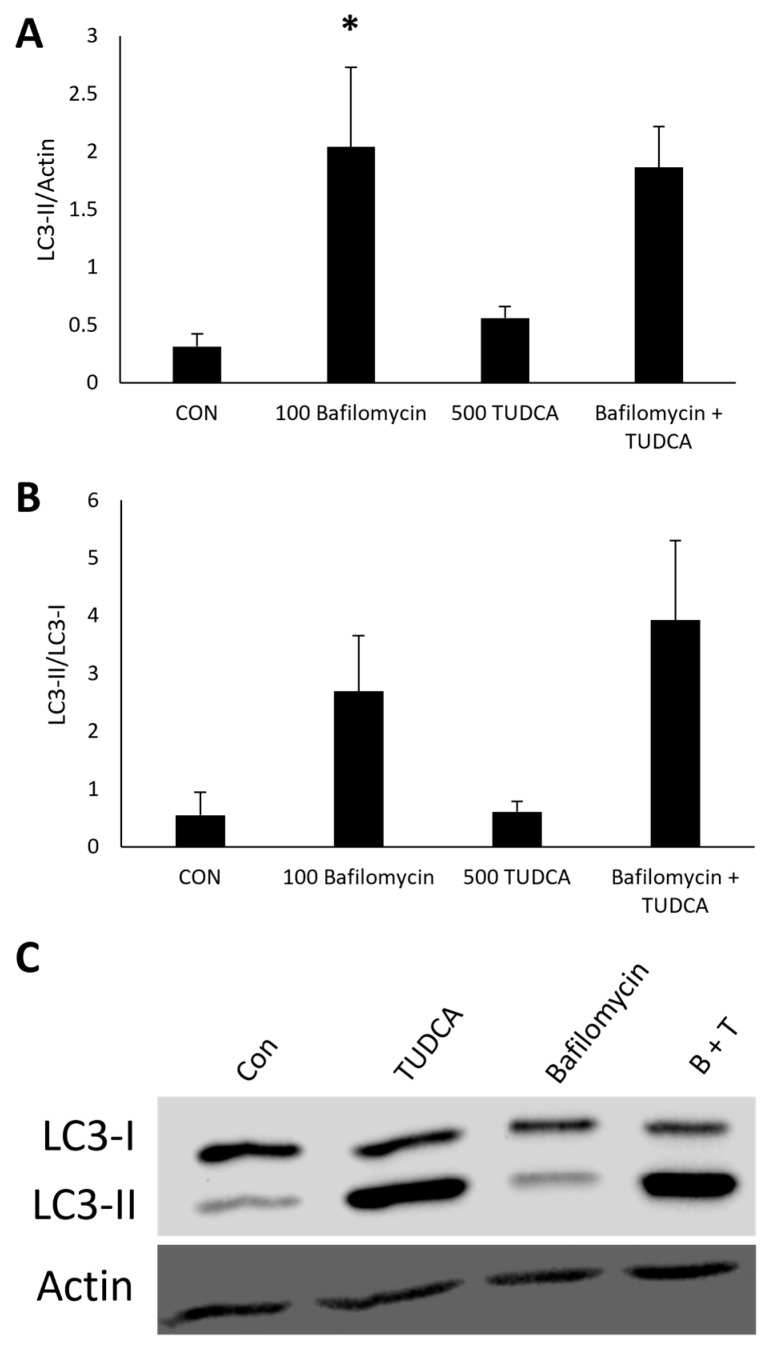
Effects of TUDCA treatment in iPSC-RPE. iPSC-RPE cells were treated with 500 µM TUDCA, 100 nM bafilomycin, or bafilomycin + TUDCA for 3 h, and protein lysate was collected to measure LC3 expression. Untreated cells served as a control. Data of LC3-II (**A**) and LC3-II/LC3-I (**B**) protein expression represented as mean ± standard deviation (n = 2). * *p* < 0.05 compared to control. (**C**) Representative Western blots of LC3.

**Figure 9 cimb-47-00224-f009:**
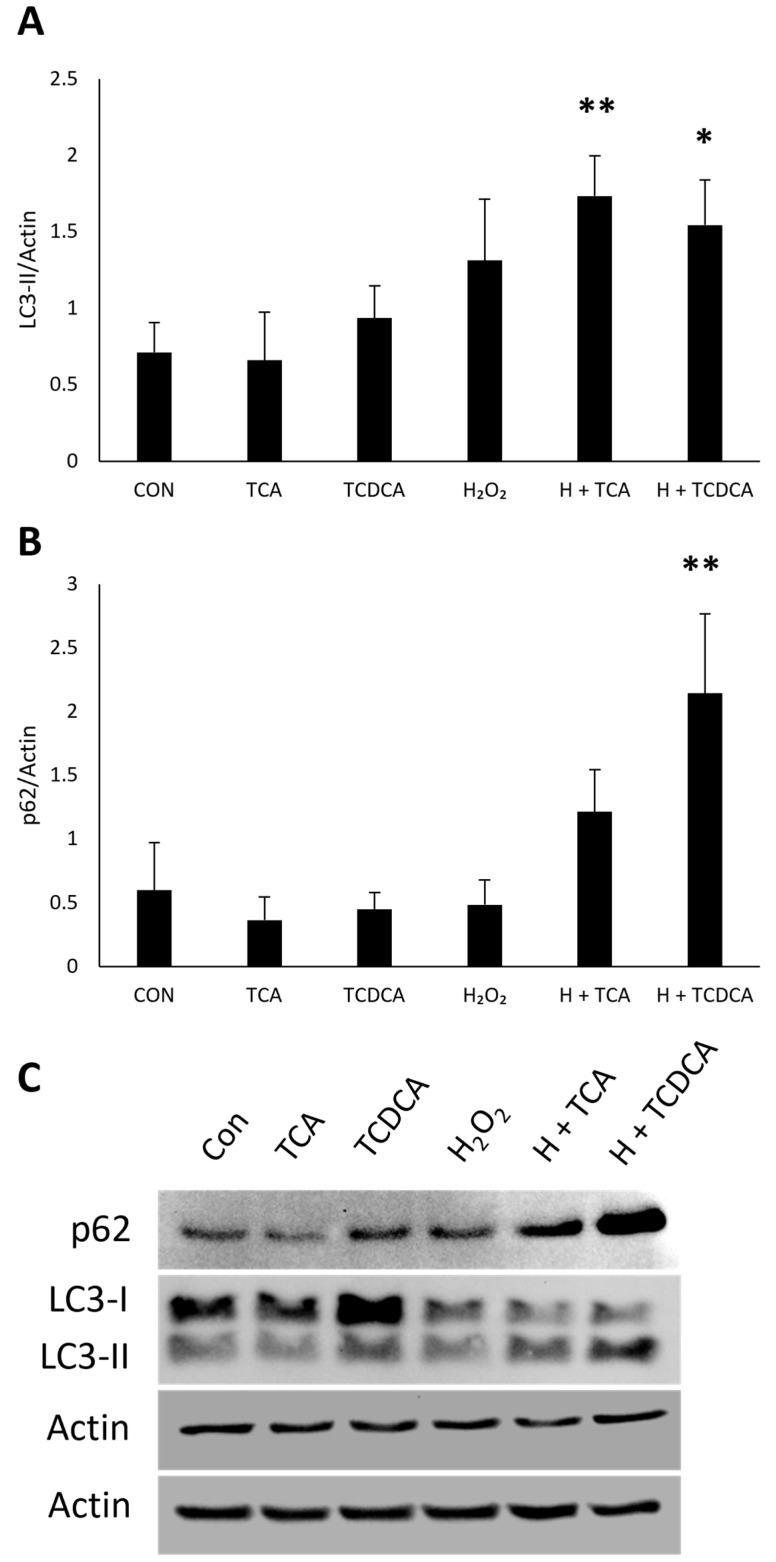
TCA and TCDCA increase LC3-II protein expression. ARPE-19 cells were treated with 500 µM TCA, 500 µM TCDCA, 800 µM H2O2, H_2_O_2_ + TCA, or H_2_O_2_ + TCDCA for 24 h, and protein lysate was collected to measure LC3 and p62 expression. Untreated cells served as a control. Data of LC3-II (**A**) and p62 (**B**) protein expression represented as mean ± standard deviation (n = 3). * *p* < 0.05 compared to control. ** *p* < 0.01 compared to control. (**C**) Representative Western blots of LC3 and p62.

## Data Availability

The raw data supporting the conclusions of this article will be made available by the authors upon request.

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
