# Peer review of "Tauroursodeoxycholic Acid Confers Protection Against Oxidative Stress via Autophagy Induction in Retinal Pigment Epithelial Cells"

_cimb, 2025, doi:10.3390/cimb47040224_

Round 1

Reviewer 1 Report

Comments and Suggestions for Authors

Summary

The authors present an interesting study on TUDCA ameliorating oxidative damage to RPE cells, which has significance to researchers interested in AMD and aging. The work uses two different RPE cell culture models (ARPE19 cells and iPSC-derived RPE cells) where they present a protective effect of TUDCA on hydrogen peroxide-induced cellular death. The authors ascribe this to an induction of autophagy. However, it is unclear whether the protective nature of TUDCA is due to an induction of autophagy. The experiments presented in this manuscript show only correlative changes, which could be resolved by additional experiments that indicate inhibiting autophagy in H202- and TUDCA-treated ARPE19 cells lessens the protective effect of the TUDCA treatment.

A major issue of this reviewer is whether the bile acid receptors are present on human RPE cells. Most of the work in this manuscript is on ARPE19 cells, which is a cell line that has not faithfully recapitulated characteristics of human RPE cells unless cultured under special conditions. In contrast, it only appears that the MCR receptor is the only one present on the iPSC-derived RPE. Is it known for TUDCA to act through this receptor? If not, then this work may not be translated to humans. Additional experiments can resolve this issue as well, which would provide a potential novel therapeutic strategy for dry AMD.

Introduction:

While it is true that a lack of appropriate animal models accurately mimicking key hallmarks of atrophic AMD complicates the development of therapeutics for dry AMD, there is no gold standard model for dry AMD including cell culture models. This statement should be revised to reflect the difficult nature of recapitulating dry AMD in any model.

RPE cells are sometimes referred to as ‘the liver of the eye.’ Do these cells make bile acids as well?

What is the receptor of TUDCA? Is there one?

Materials and Methods:

Was a positive control (i.e. a cell or tissue with known expression of the bile acid receptors) used in these studies to evaluate the presence of these receptors in the ARPE19 and iPSC-derived RPE?

Results:

This reviewer would argue that a Ct = 52.042 may suggest that TGR5 is not present in IPSC-derived RPE cells.  What were the Cts for the housekeeping gene used in this study? What is the Ct threshold detection limit of the authors?

Did the authors examine cell lysates from iPSC-derived RPE for the bile acid receptors? This should be performed and included to determine if these receptors are present in non-proliferative polarized RPE cells that are more closely mirroring RPE in vivo than ARPE19 cells.  

The western blots for the bile acid receptors in Panel B in Figure 1 are not convincing. There are many bands for PXR, and there are multiple bands for TGR5 and FXR (none of which match the expected size provided by the manufacturer’s website). Can the authors perform additional experiments to provide more compelling data that these receptors are present on RPE cells? This is critical for this study because if these receptors are not present, then how can TDUCA possibly influence RPE homeostasis?

Can the authors describe the discrepancy between the LC3II (normal relative to control in H202 treated and TUDCA treated) and P62 results (decreased relative to control in H202 treated and TUDCA treated) in Figure 3?  

Can the authors describe what an n represents in this study (i.e in Figure 3,5-7)?

If the legend for Figure 3 is correct, LC3 in Panel A should be LC3II.

Can the authors provide higher magnification images for Panel A of Figure 4/Figure 8? It looks like the p62 staining is diffuse and not puncta related. What is the magnification of these images? Scale bars? These cells also look like they are not confluent but quite sparse.

According to Figure 3, I would anticipate the LC3/p62 ratio at distinct puncta in either H202- or TUDCA-treated cells to be increased while the H202 and TUDCA treated-cells to be the highest in Figure 4. Can the authors provide an explanation why this isn’t the case? Is it the possibility that the LC3 antibody does not differentiate between the LC3BI and LC3BII forms?

The western blot in Figure 5 does not match the results in the graph in Panel B.

The levels of Akt and p-Akt are very low in the H202 and TUDCA-treated cells in Figure 7. Are there other markers for the activity of mTORC2 that can be examined?

Although under the right conditions, ARPE19 cells can resemble RPE cells in vivo, the ones in this study do not (see Figure 4). The authors may want to revise the introductory sentence to Section 3.6.

In Figure 8, the images in Panel A do not resemble RPE cells. Did the authors confirm that these cells are resembling RPE cells by labeling their tight junctions?

The data provided in Figure 9 does not show that TUDCA increases autophagic flux. The authors may want to revise the figure legend title to reflect this. The reviewer can appreciate the authors’ explanations on why they could have obtained these results (i.e. small sample sizes/not optimized differentiation protocols); however, this raises some concern, especially in regard to the differentiation protocol. What was not optimized in this protocol? Do the authors think the results obtained in this study would be different if the cells were treated differently?

In Figure 10, it would have been nice to include TUDCA as a treatment for these experiments so the effects of TUDCA could be compared to TCA and TCDCA. I am unsure if this figure is needed in this manuscript because the manuscript is focused on TUDCA.

Author Response

For: Tauroursodeoxycholic Acid Confers Protection Against Oxidative Stress via Autophagy Induction in Retinal Pigment Epithelial Cells

Reviewer 1: The authors present an interesting study on TUDCA ameliorating oxidative damage to RPE cells, which has significance to researchers interested in AMD and aging. The work uses two different RPE cell culture models (ARPE19 cells and iPSC-derived RPE cells) where they present a protective effect of TUDCA on hydrogen peroxide-induced cellular death. The authors ascribe this to an induction of autophagy. However, it is unclear whether the protective nature of TUDCA is due to an induction of autophagy. The experiments presented in this manuscript show only correlative changes, which could be resolved by additional experiments that indicate inhibiting autophagy in H202- and TUDCA-treated ARPE19 cells lessens the protective effect of the TUDCA treatment.

We thank the reviewer and appreciate their careful evaluation of our study. The study demonstrates TUDCA’s protective effect against Hâ‚‚Oâ‚‚-induced oxidative damage in RPE cells, suggesting a correlation with autophagy induction. To clarify whether autophagy mediates this protection, experiments using autophagy inhibitors, such as 3-MA or chloroquine, could determine if blocking autophagy reduces TUDCA's efficacy. However, due to limitations in time and resources, it is not feasible to confirm causation through further experiments at this stage. We acknowledge this constraint and propose that future studies investigate whether inhibiting autophagy impacts TUDCA’s protective role. This mechanistic understanding could provide deeper insights into TUDCA’s therapeutic potential for AMD and aging-related retinal conditions. This explanation was added to our manuscript at Lines 483-489.

Comment 1: A major issue of this reviewer is whether the bile acid receptors are present on human RPE cells. Most of the work in this manuscript is on ARPE19 cells, which is a cell line that has not faithfully recapitulated characteristics of human RPE cells unless cultured under special conditions. In contrast, it only appears that the MCR receptor is the only one present on the iPSC-derived RPE. Is it known for TUDCA to act through this receptor? If not, then this work may not be translated to humans. Additional experiments can resolve this issue as well, which would provide a potential novel therapeutic strategy for dry AMD.

Response 1: The reviewer comments on the presence of bile acid receptors in RPE cells. Daruich et al. (2019) provide a clear table of various bile acids with their respective bile acid receptors and location in humans (Table 1). For the MCR receptor, we can see that TUDCA has an affinity to act through this receptor and that this receptor is located in the retina. Therefore, our work has the potential to be translated to humans. We agree that additional experiments would be great next steps to provide further evidence that these receptors are present in human RPE cells; however, we do not have the funds or the resources to perform these additional experiments now.

We cite the paper by Daruich et al. in our manuscript (reference 50), and the PMID is here: https://pubmed.ncbi.nlm.nih.gov/31700226/.

Introduction:

Comment 2: While it is true that a lack of appropriate animal models accurately mimicking key hallmarks of atrophic AMD complicates the development of therapeutics for dry AMD, there is no gold standard model for dry AMD including cell culture models. This statement should be revised to reflect the difficult nature of recapitulating dry AMD in any model.

Response 2: Thank you for your suggestion. We have included a sentence to indicate that there are currently no cell or animal models that accurately replicate the key aspects of age-related macular degeneration (AMD) pathophysiology. This limitation is due to the multifactorial nature of the condition, which complicates our understanding of the mechanisms involved in AMD pathogenesis. This information was added to our manuscript at Lines 73-76.

Comment 3: RPE cells are sometimes referred to as ‘the liver of the eye.’ Do these cells make bile acids as well?

Response 3: While RPE cells contain bile acid receptors with an affinity for various bile acids, the cells do not produce bile acids. The enzymes used to make bile acids in the hepatic pathways are unique to the liver. Gut microbiota produces bile acids through several processes, including hydroxylation, deconjugation, epimerization, and oxidation. We addressed this in the Introduction section of our manuscript in Lines 41-48.

Comment 4: What is the receptor of TUDCA? Is there one?

Response 4: Several receptors have an affinity for TUDCA. Daruich et al. (2019) provide a table of various bile acids and their respective bile acid receptors (Table 1). Some of the membrane receptors mentioned in this paper are TGR5, S1PR2, α5β1 integrin, and MCR. We specifically chose to measure the expression of TGR5 and MCR because these are known to be found in retinal cells.

We cite the paper by Daruich et al. in our manuscript (reference 50), and the PMID is here: https://pubmed.ncbi.nlm.nih.gov/31700226/.

Materials and Methods:

Comment 5: Was a positive control (i.e. a cell or tissue with known expression of the bile acid receptors) used in these studies to evaluate the presence of these receptors in the ARPE19 and iPSC-derived RPE?

Response 5: The reviewer’s comment that we should use a positive control to evaluate the presence of these receptors in our cells is a great idea. We agree that this would strengthen our results by providing a control to compare our current data. We should consider doing this in the future, but at this time, we do not have the funds or the resources to perform these additional experiments.

Results:

Comment 6: This reviewer would argue that a Ct = 52.042 may suggest that TGR5 is not present in IPSC-derived RPE cells.  What were the Cts for the housekeeping gene used in this study? What is the Ct threshold detection limit of the authors?

Response 6: We used TBP as an internal control for the PCR with a Ct of about 31. We extended the cycles to find low detection of the receptors. Still, we followed the widely accepted principle that a Ct value of 35 or higher is considered very low or essentially non-existent. Given that we observed a trace expression of some receptors but not others, we have chosen to report the findings for TGR5 despite its expression being at a very low level. Additionally, this study established no specific threshold detection limits for receptor expression.

Comment 7: Did the authors examine cell lysates from iPSC-derived RPE for the bile acid receptors? This should be performed and included to determine if these receptors are present in non-proliferative polarized RPE cells that are more closely mirroring RPE in vivo than ARPE19 cells. 

Response 7: We used cell lysates from iPSC-derived RPE to measure the baseline gene expression of these receptors using qRT-PCR. We did not use them to detect the relative protein expression of bile acid receptors using western blots. While we agree this would make a great addition to your manuscript, we do not have the funds or the resources to perform these experiments.

Comment 8: The western blots for the bile acid receptors in Panel B in Figure 1 are not convincing. There are many bands for PXR, and there are multiple bands for TGR5 and FXR (none of which match the expected size provided by the manufacturer’s website). Can the authors perform additional experiments to provide more compelling data that these receptors are present on RPE cells? This is critical for this study because if these receptors are not present, then how can TDUCA possibly influence RPE homeostasis?

Response 8: The reviewer comments on the western blots on the bile acid receptors we analyzed. We have added FXR and PXR immunocytochemistry staining of ARPE-19 cells to provide further evidence that these receptors are present in RPE cells (updated Figure 1). We agree that these additional images will reinforce the main points that these receptors are present in RPE cells.

Comment 9: Can the authors describe the discrepancy between the LC3II (normal relative to control in H202 treated and TUDCA treated) and P62 results (decreased relative to control in H202 treated and TUDCA treated) in Figure 3? 

Response 9: We appreciate the reviewer's comments regarding the discrepancy between the LC3-II and p62 results in Figure 3. While it may seem contradictory, it is important to note that LC3-II and p62 measure different aspects of autophagic activity and should not be compared directly. p62 is an autophagic cargo receptor degraded through the autophagic-lysosomal pathway, so its levels are inversely related to autophagic flux. A decrease in p62, as seen in our H2O2 and TUDCA-treated cells, suggests enhanced autophagic degradation. On the other hand, LC3-II is a marker of autophagosomal membranes, where LC3-I is lipidated to form LC3-II during autophagy induction. Therefore, increased LC3-II levels in these treated cells indicate active autophagosome formation. The observed differences in p62 and LC3-II reflect the distinct stages of autophagy they represent—p62 degradation versus autophagosome formation—and are not contradictory when considered in this context. This is explained in our manuscript in Lines 224-230.

Comment 10: Can the authors describe what an n represents in this study (i.e in Figure 3,5-7)?

Response 10: We used n to represent the number of independent experiments used in the data analysis. For example, in Figure 3, LC3-II has an n = 3, which means we used three replicate western blot experiments to analyze this data set.

Comment 11: If the legend for Figure 3 is correct, LC3 in Panel A should be LC3II.

Response 11: Thank you for your suggestion. We have edited the y-axis of Panel A to reflect LC3-II.

Comment 12: Can the authors provide higher magnification images for Panel A of Figure 4/Figure 8? It looks like the p62 staining is diffuse and not puncta related. What is the magnification of these images? Scale bars? These cells also look like they are not confluent but quite sparse.

Response 12: Thank you for the suggestions. We have added a scale bar to these images. The magnification of these images is 60x. We want to keep the magnification the same for all the images to avoid confusing our readers. We did not grow our cells to full confluency; instead, they were seeded onto the slides at 50% confluence. Sparse cells were used because they can provide spatial and morphological details about autophagosomes, lysosomes, and their co-localization at the single-cell level. This experimental approach is discussed in Section 2.5 of the Materials and Methods section, at Lines 155-158.

Comment 13: According to Figure 3, I would anticipate the LC3/p62 ratio at distinct puncta in either H202- or TUDCA-treated cells to be increased while the H202 and TUDCA treated-cells to be the highest in Figure 4. Can the authors provide an explanation why this isn’t the case? Is it the possibility that the LC3 antibody does not differentiate between the LC3BI and LC3BII forms?

Response 13: The reviewer comments on the LC3 antibody’s ability to differentiate between the LC3-I and LC3-II forms. The western blot images in Figure 3 show that the LC3 antibody does distinguish between the LC3-I and LC3-II forms. The same LC3 antibody was used for the immunocytochemistry shown in Figure 4. A possible explanation for this discrepancy is that the LC3 antibody used in immunocytochemistry may not effectively distinguish between LC3-I and LC3-II. Since autophagic flux increases in TUDCA-treated cells, LC3-positive puncta may not accumulate due to efficient autophagosome clearance via lysosomal degradation. Therefore, differences in autophagic flux rather than LC3 expression alone may account for the observed unexpected LC3/p62 ratio patterns.

Comment 14: The western blot in Figure 5 does not match the results in the graph in Panel B.

Response 14: The reviewer comments on the western blot image in Figure 5. We chose these western blot images specifically because they are the most representative of the data’s quantification.

Comment 15: The levels of Akt and p-Akt are very low in the H202 and TUDCA-treated cells in Figure 7. Are there other markers for the activity of mTORC2 that can be examined?

Response 15: We chose to measure Akt and p-Akt since these are markers for the activity of mTORC2. Other markers could be used, which could be great next steps in this line of research. However, we do not have the funds or resources to perform additional experiments now. We still believe that our current western blot experiments have strongly justified our conclusion.

Comment 16: Although under the right conditions, ARPE19 cells can resemble RPE cells in vivo, the ones in this study do not (see Figure 4). The authors may want to revise the introductory sentence to Section 3.6.

Response 16: Thank you for the suggestion. We have revised this sentence in our manuscript at Lines 342-345.

Comment 17: In Figure 8, the images in Panel A do not resemble RPE cells. Did the authors confirm that these cells are resembling RPE cells by labeling their tight junctions?

Response 17: The reviewer comments on the resemblance of the cells in Panel A of Figure 8 to RPE cells. We did not label the tight junctions of the iPSC-RPE, partly because we did not grow our cells to a 100% confluent monolayer. We could consider doing in the future, but at this time, we do not have the funds or the resources to repeat this experiment.

Comment 18: The data provided in Figure 9 does not show that TUDCA increases autophagic flux. The authors may want to revise the figure legend title to reflect this.

Response 18: Thank you for your suggestion. We have edited the figure legend title to state “Effect of TUDCA treatment in iPSC-RPE.” We acknowledge that while statistical significance was not reached in this dataset using iPSC-RPE cells, the biological relevance of the observed pattern remains strong, suggesting that future work with increased sample sizes or optimized differentiation protocols may help clarify this further.

Comment 19: The reviewer can appreciate the authors’ explanations on why they could have obtained these results (i.e. small sample sizes/not optimized differentiation protocols); however, this raises some concern, especially in regard to the differentiation protocol. What was not optimized in this protocol? Do the authors think the results obtained in this study would be different if the cells were treated differently?

Response 19: We appreciate the reviewer's concerns regarding the differentiation protocol. The iPSC-RPE cells used in this study were commercially sourced, meaning we did not have direct control over the differentiation process, as each vendor follows its own established protocol. While we acknowledge that differentiation variability could influence experimental outcomes, purchasing iPSC-RPE cells was the most practical approach for our laboratory, ensuring consistency and reproducibility across experiments. Nonetheless, future studies could explore alternative sources or in-house differentiation to assess potential variations in cellular responses under different differentiation conditions.

Comment 20: In Figure 10, it would have been nice to include TUDCA as a treatment for these experiments so the effects of TUDCA could be compared to TCA and TCDCA. I am unsure if this figure is needed in this manuscript because the manuscript is focused on TUDCA.

Response 20: The reviewer comments on the inclusion of TCA and TCDCA. While this manuscript focuses on TUDCA, we thought that including data with TCA and TCDCA that do not show the same autophagy induction outcomes would support the hypothesis that autophagy induction is unique to TUDCA. We know we cannot make any definitive conclusions regarding this statement without completing autophagy flux experiments using those additional bile acids. However, we hope to inspire new research with this publication that will take these next steps to build on the work we have done in the current study.

Reviewer 2 Report

Comments and Suggestions for Authors

This manuscript investigates the protective role of tauroursodeoxycholic acid (TUDCA) against oxidative stress in retinal pigment epithelial (RPE) cells, with a particular focus on autophagy induction as the underlying mechanism. The authors present evidence that TUDCA activates autophagy in RPE cells via mTORC1/mTORC2-independent but Atg5-dependent pathways, thereby protecting these cells against oxidative damage. Given the critical role of RPE dysfunction in age-related macular degeneration (AMD), this study offers valuable insights into potential therapeutic strategies for treating conditions like geographic atrophy.

Scientific Significance: The study addresses a significant gap in understanding how bile acids, particularly TUDCA, protect RPE cells from oxidative stress—a key factor in AMD pathogenesis. The findings have potential translational implications for developing novel therapies for AMD.

Methodological Approach: The authors employ complementary techniques (Western blot, immunocytochemistry, MTT assays) to investigate autophagy induction. The use of both immortalized ARPE-19 and iPSC-derived RPE cells strengthens the reliability of their findings.

Experimental Design: The experiments systematically explore TUDCA's effects on RPE cells, from establishing the expression of bile acid receptors to characterizing autophagy induction and elucidating the underlying signaling pathways. The use of bafilomycin to measure autophagy flux is particularly commendable.

Comments

Cell Line Limitations: While the authors acknowledge differences between ARPE-19 and iPSC-derived RPE cells, some autophagy flux experiments showed trends that didn't reach statistical significance in iPSC-derived RPE cells. This discrepancy warrants further discussion regarding the limitations of these cell models and potential physiological implications.

Mechanism Characterization: The study concludes that TUDCA induces autophagy via mTORC1/mTORC2-independent but Atg5-dependent pathways. However, the precise molecular mechanism linking TUDCA to Atg5 activation remains unclear. Additional experiments investigating the direct targets of TUDCA could strengthen this aspect.

Bile Acid Receptor Specificity: The authors demonstrate the expression of bile acid receptors in RPE cells but don't fully explore which specific receptor(s) mediate TUDCA's effects. Receptor antagonist/inhibitor experiments or siRNA knockdown studies would provide valuable insights into the receptor-dependent mechanisms.

Comparison with Other Bile Acids: The authors began exploring autophagy induction by other bile acids (TCA, TCDCA) but didn't complete autophagy flux experiments with these compounds. This limits the conclusions about the uniqueness of TUDCA's effects compared to other bile acids.

Recommendations

Consider Additional Mechanistic Experiments: Complete autophagy flux experiments with TCA and TCDCA to determine if autophagy induction is unique to TUDCA

Strengthen iPSC-RPE Data: Discuss potential reasons for differences between ARPE-19 and iPSC-RPE responses

Enhance Physiological Relevance: Discuss how the concentrations of TUDCA used compare to physiologically achievable levels

Conclusions

This manuscript has valuable insights into TUDCA's protective mechanism in RPE cells, highlighting its role in autophagy induction. The work is scientifically sound but needs further revisions before consideration for publication.

Author Response

For: Tauroursodeoxycholic Acid Confers Protection Against Oxidative Stress via Autophagy Induction in Retinal Pigment Epithelial Cells

Reviewer 2: This manuscript investigates the protective role of tauroursodeoxycholic acid (TUDCA) against oxidative stress in retinal pigment epithelial (RPE) cells, with a particular  focus on autophagy induction as the underlying mechanism. The authors present evidence that TUDCA activates autophagy in RPE cells via mTORC1/mTORC2-independent but Atg5-dependent pathways, thereby protecting these cells against oxidative damage. Given the critical role of RPE dysfunction in age-related macular degeneration (AMD), this study offers valuable insights into potential therapeutic strategies for treating conditions like geographic atrophy.

Scientific Significance: The study addresses a significant gap in understanding how bile acids, particularly TUDCA, protect RPE cells from oxidative stress—a key factor in AMD pathogenesis. The findings have potential translational implications for developing novel therapies for AMD.

Methodological Approach: The authors employ complementary techniques (Western blot, immunocytochemistry, MTT assays) to investigate autophagy induction. The use of both immortalized ARPE-19 and iPSC-derived RPE cells strengthens the reliability of their findings.

Experimental Design: The experiments systematically explore TUDCA's effects on RPE cells, from establishing the expression of bile acid receptors to characterizing autophagy induction and elucidating the underlying signaling pathways. The use of bafilomycin to measure autophagy flux is particularly commendable.

Comments

Comment 1: Cell Line Limitations: While the authors acknowledge differences between ARPE-19 and iPSC-derived RPE cells, some autophagy flux experiments showed trends that didn't reach statistical significance in iPSC-derived RPE cells. This discrepancy warrants further discussion regarding the limitations of these cell models and potential physiological implications.

Response 1: Thank you for your comment to add further discussion regarding the limitations of these cell models and potential physiological implications. We address the differences between the cell types in the Discussion section of the manuscript at Lines 408-411. We have also expanded on discussing of the limitations with these cell types in the manuscript at Lines 411-420.

Comment 2: Mechanism Characterization: The study concludes that TUDCA induces autophagy via mTORC1/mTORC2-independent but Atg5-dependent pathways. However, the precise molecular mechanism linking TUDCA to Atg5 activation remains unclear. Additional experiments investigating the direct targets of TUDCA could strengthen this aspect.

Response 2: The reviewer suggests using additional experiments to investigate the direct targets of TUDCA to strengthen our conclusion that TUDCA induces autophagy via Atg5-dependent pathways. While we agree that these types of experiments would be great next steps in this line of research, this is beyond the scope of the current study. We do not have the funds or the resources to be able to perform additional experiments at this time. We still believe we have conveyed a strong rationale for our conclusion with our current western blot experiments.

Comment 3: Bile Acid Receptor Specificity: The authors demonstrate the expression of bile acid receptors in RPE cells but don't fully explore which specific receptor(s) mediate TUDCA's effects. Receptor antagonist/inhibitor experiments or siRNA knockdown studies would provide valuable insights into the receptor-dependent mechanisms.

Response 3: The reviewer suggests using additional experiments to investigate the receptor-dependent mechanisms of TUDCA. While we agree that these types of experiments would be great next steps in this line of research, they are beyond the scope of the current study. Moreover, we do not have the funds or the resources to be able to perform additional experiments now.

Comment 4: Comparison with Other Bile Acids: The authors began exploring autophagy induction by other bile acids (TCA, TCDCA) but didn't complete autophagy flux experiments with these compounds. This limits the conclusions about the uniqueness of TUDCA's effects compared to other bile acids.

Response 4: The reviewer comments on the limitations of not completing autophagy flux experiments with TCA and TCDCA. We wanted to introduce the idea that autophagy induction was unique to TUDCA, but we know we cannot make this a definitive conclusion without completing autophagy flux experiments with other bile acids. We address this limitation in the Discussion section of the manuscript at Lines 541-545.

Recommendations

Recommendation 1: Consider Additional Mechanistic Experiments: Complete autophagy flux experiments with TCA and TCDCA to determine if autophagy induction is unique to TUDCA

Response 1: We appreciate the reviewer's suggestion to investigate autophagy induction with TCA and TCDCA. We agree this would be a valuable next step in determining whether autophagy induction is unique to TUDCA. However, as previously noted, due to constraints in funding, time, and resources, we are unable to conduct these additional experiments within the current study. Given the 10-day revision period, we hope our findings will serve as a foundation for future research to explore this aspect in more detail. We believe our current data offer significant insight into the autophagic effects of TUDCA, and we are confident that follow-up studies will expand on this work.

Recommendation 2: Strengthen iPSC-RPE Data: Discuss potential reasons for differences between ARPE-19 and iPSC-RPE responses

Response 2: The reviewer suggests we discuss potential reasons for differences between ARPE-19 and iPSC-RPE responses. We addressed these differences in the Discussion section of the manuscript at Lines 408-411 and expanded on them in the manuscript at Lines 411-420.

Recommendation 3: Enhance Physiological Relevance: Discuss how the concentrations of TUDCA used compare to physiologically achievable levels

Response 3: Thank you for your recommendation to discuss the concentrations of TUDCA used in our experiments. In previous work, we used a dose curve of the bile acid ranging from 10 to 500 μM and found that RPE cells exhibited improved protection with increasing doses. Protection plateaued at higher doses (200-500 μM), leading us to select 500 μM to ensure optimal protective effects in this study. We have added this information to the manuscript at Lines 99-100.

The previous work we refer to are Warden et al. (2020) and Warden and Brantley (2021), https://doi.org/10.1016/j.exer.2020.107974 and https://doi.org/10.3390/biom11050626, respectively.

However, to address physiological relevance, it is important to note that the concentrations used in vitro often exceed those that are achievable in vivo due to differences in cellular exposure and metabolic processing. TUDCA levels in human plasma are typically reported in the nanomolar to low micromolar range under physiological conditions, even when supplemented. Therefore, while our findings demonstrate the potential protective effects of TUDCA at higher concentrations, further studies are necessary to translate these results to physiologically achievable levels and to evaluate their effectiveness in vivo.

Conclusions

This manuscript has valuable insights into TUDCA's protective mechanism in RPE cells, highlighting its role in autophagy induction. The work is scientifically sound but needs further revisions before consideration for publication.

Thank you for your comments and recommendations for our manuscript. We have implemented many of your suggestions in this revised submission, and we agree that these revisions have made our work more valuable.

Round 2

Reviewer 1 Report

Comments and Suggestions for Authors

The authors have carefully revised their manuscript according to my suggestions. With the addition of the new reference (PMID: 31700226), this reviewer's concerns about the expression of these receptors on RPE cells is lessen. However, this reviewer would recommend the authors to consider removing Figure 1 from their manuscript given this reference, which is still not convincing. 

This reviewer also appreciates the additional information in the materials and methods section of the manuscript. However, this reviewer would recommend for future studies that RPE cells, especially iPSC-derived RPE cultures, be grown to confluence. This more accurately reflects the true nature of these cells. 

Author Response

Comment 1: The authors have carefully revised their manuscript according to my suggestions. With the addition of the new reference (PMID: 31700226), this reviewer's concerns about the expression of these receptors on RPE cells is lessen. However, this reviewer would recommend the authors to consider removing Figure 1 from their manuscript given this reference, which is still not convincing.  

Response 1: The reviewer recommends removing Figure 1 from our manuscript. The information presented in this figure is new and acts as an important foundation for the remainder of the experiments in this study. Based on the reviewer’s suggestion, we removed Figure 1 from the main manuscript and included it as a supplementary figure.  

Comment 2: This reviewer also appreciates the additional information in the materials and methods section of the manuscript. However, this reviewer would recommend for future studies that RPE cells, especially iPSC-derived RPE cultures, be grown to confluence. This more accurately reflects the true nature of these cells.  

Response 2: The reviewer recommends that RPE cells, especially iPSC-derived RPE cells, be grown to confluence for future studies. We appreciate the recommendation and will apply it to our future experiments. Our goal is to accurately reflect the true nature of the cells, so it will be important to follow the reviewer’s recommendation to grow them to confluence.  

Reviewer 2 Report

Comments and Suggestions for Authors

The manuscript now presents a clearer picture of TUDCA's protective effects on RPE cells through autophagy induction, with appropriate acknowledgment of limitations and areas for future research. The study makes a significant contribution to understanding potential therapeutic approaches for age-related macular degeneration, particularly for geographic atrophy which currently has limited treatment options.

Author Response

Comment 1: The manuscript now presents a clearer picture of TUDCA's protective effects on RPE cells through autophagy induction, with appropriate acknowledgment of limitations and areas for future research. The study makes a significant contribution to understanding potential therapeutic approaches for age-related macular degeneration, particularly for geographic atrophy which currently has limited treatment options. 

Response 1: We appreciate the reviewer's comments and recommendations on our manuscript. They have strengthened our study and made it more valuable for the vision research community.